

# Exploring the Tidal Response to Bathymetry Evolution and Present-Day Sea Level Rise in a Channel-Shoal Environment

Robert Lepper[1], Leon Jänicke[2], Ingo Hache[1], Christian Jordan[3], and Frank Kösters[1]

[1]Federal Waterways Engineering and Research Institute (BAW), Hamburg, 22880, Germany
[2]IU-International University of Applied Sciences, Duisburg, 47059, Germany
[3]Leibniz University Hannover, Ludwig-Franzius Institute of Hydraulic, Estuarine and Coastal Engineering, Hanover, 30060, Germany

*Correspondence to*: Robert Lepper (robert.lepper@baw.de), ORCID: 0000-0002-8446-2004

**Abstract.** Intertidal flats and salt marshes in channel-shoal environments are at severe risk from drowning under sea level rise (SLR) ultimately ceasing their function in coastal defense. Earlier studies indicated that these environments can be resilient against moderate SLR as their mean height is believed to correlate with tidal amplitude and mean sea level. Recent morphological analyses in the German Wadden Sea on the Northwestern European Shelf contradicted this assumption as mean tidal flat accretion surpassed relative SLR; indicating that nonlinear feedback between SLR, coastal morphodynamics, and tidal dynamics played a role. We explored this relationship in the German Wadden Sea's channel-shoal environment by revisiting the sensitivity of tidal dynamics to observed SLR and coastal bathymetry evolution over one nodal cycle (1997 to 2015) with a numerical model. We found a proportional response of tidal high and low water to SLR when the bathymetry was kept constant. In contrast, coastal bathymetry evolution caused a spatially-varying hydrodynamic reaction with both increases and decreases of tidal characteristic patterns within few kilometers. An explorative assessment of potential mechanisms suggested that energy dissipation declined near the coast which we related to decreasing tidal prism and declining tidal energy import. Our study stresses the fact that an accurate representation of coastal morphology in hind- and nowcasts and ensembles for bathymetry evolution to assess the impact of SLR are needed when using numerical models.

## 1 Introduction

Channel-shoal environments with extensive intertidal flats develop whenever the tidal range is large in comparison to the significant wave height (Hayes, 1979). Shallow bed slopes and low water depths during tidal flooding serve, for example, as natural coastal protection measures with low maintenance, natural carbon sinks, or unique habitats (Stive et al., 1990). This makes coastal channel-shoal morphologies an efficient, low-cost energy dissipation zone in front of coastal defense structures (Möller et al., 2014). Future global mean sea level rise (SLR) of 0.63 to 1.01 m (SSP5-8.5) until the end of the 21st century compared to the reference period of 1995 to 2014 (Fox-Kemper et al., 2021) will have a significant but yet unknown effect on these environments. The key question is if these channel-shoal systems can cope with future SLR. Their loss would



make coastal protection infeasible at many locations around the world and ultimately lead to flooding and to the loss of low-lying land due to submergence.

The degree of permanent inundation is, however, not only dependent on mean sea level (MSL) but also on the co-development of tidal characteristics as tidal high water and storm surge water levels largely determine the local impact of flooding. Tidal characteristics were shown to respond nonlinearly to SLR in shallow shelf seas depending on the SLR

magnitude and on the co-evolution of the coastal morphology (Lee et al., 2017; Jordan et al., 2021; Wachler et al., 2020), which in turn is shaped by tidal and wave energy (Hofstede, 2015; Hofstede et al., 2018). This interaction results in a spatially varying impact on tidal properties such as amplitude, phase, and asymmetry which may either dampen or amplify the consequences of SLR in coastal environments even further. An assessment of sea surface heights in the German Wadden Sea in the North Sea found a linear relationship between regional SLR and increases in tidal high water. Unexpectedly, such

a correlation was absent in tidal low water development. A link to either local morphology or changes in water column stratification was suspected (Jänicke, 2021; Jänicke et al., 2020).

Numerical simulations in Chesapeake Bay, Delaware Bay, and San Francisco Bay (US) all indicated decreasing tidal amplitude if flooding of low-lying terrain (i.e., coastal hinterland, salt marshes, or tidal flats) under SLR was permitted (Holleman and Stacey, 2014; Lee et al., 2017), with similar findings in the Chinese Bohai Bay (Pelling et al., 2013b), and in

the North Sea (Pelling et al., 2013a). In these studies, a SLR-related loss of low-lying terrain and intertidal storage was connected to tidal amplification, increased flood dominance, and increased tidal flow velocity (Wachler et al., 2020). Simplified modeling studies established that SLR can cause morphodynamic adaptation ranging from accretion (Elmilady et al., 2022), to eventual drowning (Becherer et al., 2018). However, none of the latter considered the possibility of lateral intertidal expansion or accretion outpacing SLR. Consequently, SLR impact modelers still debate the flooding vs. no-

flooding option for the low-lying coastal terrain with an emphasis on tidal and wave energy and its dissipation. Recently observed increases in relative intertidal storage from such changes in the German Wadden Sea (North Sea) questioned the assumption that tidal flats will be lost under SLR (Hagen et al., 2022; Lepper, 2023). This emphasizes that a deeper understanding of the feedback between tides, SLR, and morphological changes is needed as the tidal flat height was shown to determine whether the tidal amplitude increases or decreases under SLR (Jordan et al., 2021; Pelling et al., 2013b).

We chose to revisit these interactions at the world's largest coherent channel-shoal system – the Wadden Sea on the northwestern European Continental Shelf. Here, intertidal accretion exceeded local SLR of 2.7 mm yr$^{-1}$ (Steffelbauer et al., 2022) with accretion rates of up to 2 cm yr$^{-1}$ in the period of 1996 to 2016, while also expanding laterally (Nederhoff et al., 2017; Vet et al., 2017; Benninghoff and Winter, 2019; Lepper, 2023). Interestingly external forcing such as wind (Krieger et al., 2021), waves, or storm surges (Lepper, 2023) was near-constant in this period. This unexpected accretion surplus

challenges the projected loss of intertidal flats under SLR mentioned above and motivated us to look closer into the response of tidal dynamics to bathymetry evolution under SLR. In the past, SLR and land reclamation in the Wadden Sea were both important drivers for the formation of today's bathymetry (Vos and Knol, 2015; Xu et al., 2022). Our research focuses on investigating the relationship between coastal morphodynamics, SLR, and tides in the German Wadden Sea. We assessed the





effects of observed SLR and bathymetry changes on tidal dynamics separately over the most recent nodal cycle of 1997 to
2015 in a numerical model. The following research questions were addressed: (1) To what extent has observed SLR contributed to changes in tidal characteristics? (2) What are the local and regional changes in tidal dynamics that result from recent coastal bathymetry evolution under SLR? (3) Do these processes interact with or amplify each other?

This paper structures as follows: First, we introduce our study site and the numerical modeling setup. Second, the design of the sensitivity study is explained and the relevant analysis parameters for understanding the recent phenomena are described. The results start with an overview of the recent bathymetry evolution and follow with the implications of bathymetry evolution and SLR on tidal dynamics. Finally, potential mechanisms, i.e., energy flux, energy dissipation, and tidal prism, were linked with the observed changes in water volume to explain the nonlinear changes in tidal characteristics.

## 2 Material and methods

### 2.1 The German Wadden Sea

The German Wadden Sea is a vital sediment and carbon sink (Stive et al., 1990) with a coastline of about 450 km, water depths of less than 50 m, and a surface area of roughly 11,000 km² (Figure 1). It consists mainly of broad, poorly-vegetated, or bare tidal flats. The flats are breached by deep tidal channels and several estuaries, tidal inlets with barrier islands, salt marshes, and shoals with shallow embankments. Its morphology was shaped by tide-dominated to mixed-energy conditions (Herrling and Winter, 2014) with an average mesotidal range of 2.8 m (Jänicke, 2021), and frequent storm surges between September and April. Individual tidal inlets form a sediment-sharing system consisting of a tidal inlet, an ebb tidal delta, updrift and downdrift barrier island coasts, all of which strive towards dynamical equilibrium (Elias et al., 2012; Wang et al., 2018). We divided the coastal zone in 27 subunits (i.e., morphological units) with similar tidal properties (Hagen et al., 2022), e.g., a tidal basin, in all following analyses (see Figure 1). Unit boundaries were defined approx. along the boundaries of tidal basins (i.e., along watersheds), along the flow direction of long tidal channels or estuaries, seawards at the tidal or estuarine inlet, and at the landward border of the numerical model.

### 2.2 Numerical modeling

We deployed a validated three-dimensional numerical model of the North Sea which extends from Scotland to the English Channel into the Wadden Sea and parts of the western Baltic Sea (Figure 1, a). Model specifications and validation were described in previous work (Hagen et al., 2021); only a brief summary is presented here. We used the hydrodynamic numerical modeling framework UnTRIM² and its well-established subgrid approach for detailed bathymetry and water volume representation (Casulli, 2009). The model was run with long-term averaged runoff and the initial salinity and temperature distributions were adapted from a 4-month model spin-up. Atmospheric pressure, surface temperature, and wind used the COSMO-REA6 reanalysis (Bollmeyer et al., 2015). The computational grid consists of approximately 202,000 horizontal unstructured cells, more than 10,000,000 refining subgrid cells, and 54 vertical z-layers with a minimum height of



0.5 m near the sea surface gradually becoming coarser downwards. Model bathymetry outside of the focus area (i.e., Figure 1, b) was interpolated from the Dutch Vaklodingen data and the European EMODnet digital terrain model (EMODnet Bathymetry Consortium, 2018). Inside of the focus area, 10 m bathymetry grids were merged from a multitude of observational surveys with different measurement technique, frequency, and accuracy using a specialized interpolation algorithm in time and space (Sievers et al., 2020; Sievers et al., 2021). Model validation for the year 2015 corresponded to

observations with an average RMSE of 10.6 cm in high water, 13.3 cm in tidal range, and 17.6 minutes in flood duration (Hagen et al., 2021).

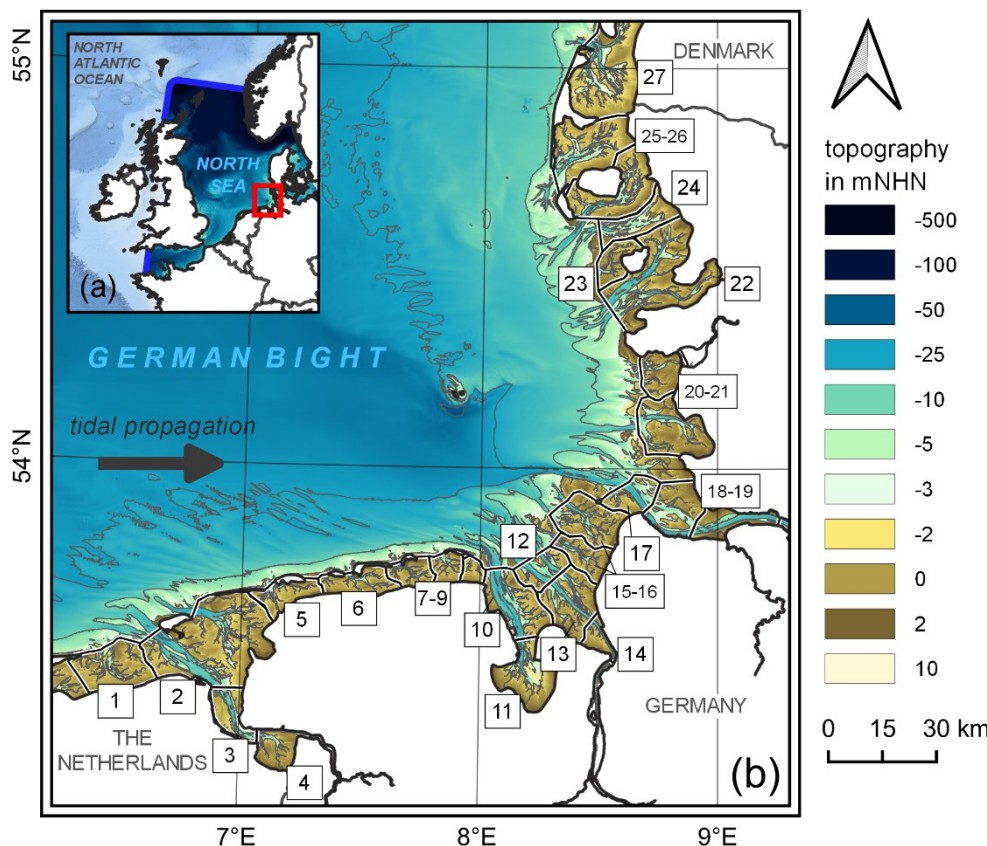

**Figure 1: Panel (a) shows the numerical model's extent on the Northwest European Continental Shelf (a). The model's open boundary is marked by a blue line and the study site (panel b) by a red rectangle. Panel (b) shows the topography of the year 2015**
**in the German Bight with brown and white patches indicating the intertidal zone and land. The tidal wave propagates counterclockwise through the German Bight. Gray contours represent the -25, -10, and -2 mNHN (with NHN being German chart datum) height isobaths. Morphological units in the coastal zone (black polygons with a number ID) were used in tidal analyses.**

### 2.3 Sensitivity experiments

Four scenarios were computed in the numerical model to derive the sensitivity of tidal characteristics to SLR and bathymetry

evolution (Table 1). All scenarios were run over a 7-month simulation period from spring to winter at the diurnal nodal



minimum of the year 2015. We chose a 7-months simulation period to represent multiple spring neap cycles. As we planned to assess the development of tidal characteristics over time, the reference scenario (1) considered the bathymetry and the mean sea level (MSL) of the year 1997. The bathymetry evolution scenario (2) used only the bathymetry, and the SLR scenario (3) only the MSL, for the year 2015. Scenario (4) used both the MSL and bathymetry of the year 2015. The

technical implementation of the scenarios is explained below. Resulting changes were always expressed forward in time, i.e., as the difference to the reference scenario. The isolated effect of bathymetry evolution, for example, is represented by the difference of (2) minus (1).

Bathymetry evolution was only considered in the morphologically active coastal zone of the German Bight (i.e., roughly within the -25 mNHN depth isobath), as bathymetry changes in the Dutch zone are still overlain by land reclamation in the

1950s (Elias et al., 2012) and Danish bathymetry were unavailable to us. All other remote and estuarine bathymetry was kept constant. It is recognized that natural morphodynamics are the result of interdependent processes (e.g., tides, waves, estuarine sediment trapping, biological potential, storm-driven erosion, etc.) which imposes a limitation to this scenario's design that cannot be mitigated. As most external forcing, aside from MSL, remained fairly constant at our study site, however, we are confident that SLR was one of the most dominant drivers behind coastal morphodynamics in the studied

nodal cycle (Lepper, 2023).

MSL in the year 2015 was estimated at -0.06 mNHN based on observational data (gauge Helgoland, Germany) and was artificially reduced by 5.9 cm (i.e., 3.2 mm yr$^{-1}$ for 18.6 yr) at all open boundaries to estimate the MSL of 1997. SLR of 3.2 mm yr$^{-1}$ was chosen considering the SLR rates from the IPCC report (Fox-Kemper et al., 2021) which corresponded with a local study that found SLR of 2.7±0.4 mm yr$^{-1}$ from observational data in the southern North Sea (Steffelbauer et al., 2022).

Using the observed MSL of the year 1997 in our model would have caused misleading bias from perennial MSL variability.

**Table 1: Overview of the numerical scenarios conducted in this study. The year 1997 represents the reference state whereas bathymetry evolution, SLR, and their combination use MSL or bathymetry of the year 2015.**

| ID | Scenario | MSL | Bathymetry |
|---|---|---|---|
| (1) | reference | 1997 | 1997 |
| (2) | bathymetry evolution | 1997 | 2015 |
| (3) | SLR | 2015 | 1997 |
| (4) | bathymetry evolution and SLR | 2015 | 2015 |

## 2.4 Tidal and bathymetry analysis

We conducted a tidal analysis of modeled sea surface heights to synthesize key tidal parameters (e.g., tidal range or tidal prism). Our analysis framework presumes a semidiurnal tidal signal to separate flood and ebb properties; therefore, we excluded all samples with low sea surface elevation variance ($\sigma < 0.1$ m) and with wetting and drying. Tidal prism was estimated using the flood water flux over the seaward-oriented transects of the morphological units (black polygons in



Figure 1, b) in tidal basins and estuaries. Water flux and tidal characteristics were averaged over the modeled period (i.e.,

409 semidiurnal tides) using tide-duration and current-duration weighted averages. The impact of our numerical experiments on tides was measured using the averaged tidal high and low water as indicators for vertical displacement, mean tidal range for tidal amplitude, and mean flood duration for tidal duration asymmetry as indicator for changes in residual sediment transport.

The barotropic energy flux and dissipation were chosen as estimators for sinks in the momentum balance of the water body

along the path of tidal propagation. The cumulative sum of pressure-related transmissive capacity and the kinetic energy loss between computational cells were computed to estimate energy fluxes and losses (Kang and Fringer, 2012). Energy dissipation was aggregated and energy fluxes were computed at the seaward boundary of morphological units (Figure 1, b). Constant tidal forcing at the open boundary minimizes the influence of other possible energy sinks, e.g., from water column stratification or tidal straining. Some changes in baroclinity cannot be avoided as MSL variation by 5.9 cm may cause bias.

Nonetheless, we assume this effect to be small considering the shallow coastal context of our study site, minor magnitudes of SLR, and strong tidal mixing.

The intertidal zone was estimated using the mean tidal high and low water data of the numerical simulation with the MSL and bathymetry of 1997 and 2015 (scenario 1 and 4, Table 1) averaged in morphological units on the 10 m bathymetry grids (Sievers et al., 2020). All bathymetry samples below mean tidal low water were assigned to the subtidal and all values in

between mean tidal high and low water to the intertidal zone. Mean tidal high and low water samples were averaged within each unit to merge the coarser modeling with finer bathymetry information. Bulk bathymetry parameters are the mean subtidal heights for subtidal deepening, mean intertidal height for intertidal accretion, and the relative intertidal area (normalized by sum of the subtidal and intertidal area) for lateral intertidal expansion.

## 3 Results

### 3.1 Bathymetry evolution

We reviewed the discrete bathymetry evolution using the height difference between the years 1997 and 2015 (Table 1, scenario 4 minus 1) at a constant reference level NHN. Changes in bulk bathymetry parameters from SLR alone were negligible (not included) and were therefore not discussed in the following. The height difference revealed complex erosion and accretion patterns. In particular, numerous small-scale channel migrations, basin-wide intertidal height changes, subtidal

retreat, and an erosion of the outer ebb-tidal deltas were observed (Figure 2, a). This visual impression was confirmed by the rightward shift of the height sample probability density function of the years 1997 and 2015 with a higher density of shallower samples between -5 and 0 mNHN in 2015 (Figure 2, b). Minor deepening was observed between -15 and -5 mNHN. Mark that unit 18 was excluded from subsequent discussion as morphological changes in this unit were biased by the natural formation of a second major ebb channel resulting in erosion and accretion of multiple meters.





**Figure 2: Panel (a) shows the difference in model bathymetry between the years 1997 and 2015, morphological units (black polygons with number ID) with accretion and erosion being represented by red and blue patches. The probability density function (PDF) of height samples between - 20 and +5 mNHN in panel (b) indicates the height distribution of the years 1997 (gray) and 2015 (blue). Differences in mean subtidal height (c), mean intertidal heights (d), and relative intertidal area (e) describe the bathymetry evolution from (a) and (b) as bulk bathymetry parameters. The left axis represents absolute values in the year 1997 and the right axis their development over time with positive values representing increases. The dashed line in (d) represents the SLR of 5.9 cm of the reference period on the right y-axis to validate the correlation between SLR and intertidal accretion. The x-axis indicates the morphological units.**

Bulk bathymetry parameters in morphological units (units indicated in Figure 2, a) confirmed an average subtidal deepening of -0.44 m in the period of 1997 to 2015 (Figure 2, c). Mean intertidal accretion (Figure 2, d) exceeded SLR more than twofold with an average increase of 0.14 m. The dashed line (Figure 2, d) presents evidence that the mean intertidal height





followed SLR in only few morphological units. Subtidal deepening and intertidal accretion were highest in the southern German Bight (units 5 to 17). The relative intertidal area (Figure 2, e, relative to the sum of the subtidal and intertidal area) expanded on average by 4.5% with peak increases of 12% in the Southwest. Increases in lateral intertidal extent often

coincided with noteworthy subtidal deepening and low intertidal accretion (e.g., units 1 to 4). Furthermore, we observed that subtidal deepening clustered in the southern German Bight (unit 2 to 17), accretion in the southeastern German Bight (unit 5 to 17), and lateral expansion in the West (unit 2, 3, and 5).

### 3.2 The sensitivity of tidal characteristics to SLR and bathymetry evolution

SLR alone (Table 1, scenario 3 minus 1) increased mean tidal high water almost equally to the prescribed SLR magnitude

(Figure 3, a). When considering bathymetry evolution without SLR (Table 1, scenario 2 minus 1), we found a spatially-varying response (Figure 3, b). The mean tidal high water remained constant in the southwestern German Bight, decreased marginally in the Southeast, and increased regionally in the Northeast. Bathymetry evolution thus introduced nonlinear tidal high water increases along hundreds of kilometers along the path of tidal propagation. Combining SLR and bathymetry evolution (Table 1, scenario 4 minus 1) yielded an overall increase in mean tidal high water with bathymetry evolution

compensating SLR-related tidal high water increases in the Southeast and amplifying them in the Northeast of the study site (Figure 3, c).

SLR also increased the mean tidal low water proportional to SLR magnitude (Figure 3, d). Contrary, bathymetry evolution decreased the mean tidal low water by several centimeters in the coastal zone with only few exceptions (Figure 3, e). For a combination of both scenarios, the proportional increase from SLR was superimposed with the spatial variability from

bathymetry evolution, resulting in either constant or amplified tidal range (Figure 3, f).

Tidal range was hardly affected by SLR, aside from minor decreases in the northeastern German Bight (Figure 3, g). As can be inferred from the tidal high and low water changes, bathymetry evolution caused a spatially-varying response in tidal range with peak increases in the Northeast of the study site and in the estuarine channels (Figure 3, h). The changes in tidal range under bathymetry evolution, reflected the changes in tidal high and low water, dominated by stronger decreases in tidal

low water. For the combined scenario, nearly all spatially-varying changes in tidal range can be attributed to local coastal bathymetry evolution rather than SLR (Figure 3, i). Note that several tidal basins in the Southeast demonstrated diminishing tidal range from bathymetry evolution which other authors linked to land reclamation in the Meldorf Bight in the 1970s (Jänicke, 2021). The impact of recent SLR on flood duration was, again, marginal, except for regional increases of less than 5 minutes in the northeastern German Bight (Figure 3, j). The development of mean flood duration under bathymetry

evolution can be split into competing regional and local phenomena (Figure 3, k). We observed regionally decreasing flood duration in the East along with local increases in tidal channels landward of barrier islands. In fact, peak increases in flood duration were noticed in nearly all tidal basins in the south and southeastern part of the German Bight. Bathymetry-related changes were again dominant with slight regional compensation of different effects at the northeastern part of the study site (Figure 3, l). In general, SLR increased whereas bathymetry evolution decreased the average flood duration.







**Figure 3: The development of the tidal high water (a-c), tidal low water (d-f), tidal range (g-i), and flood duration (j-l). Changes associated with observed SLR, bathymetry evolution, and their combination are shown in the left, middle, and right columns, respectively. Red and blue patches indicate an increase and a decrease of the respective tidal characteristic. White patches represent no-data values or land while gray patches indicate the intertidal zone at low water. Note that all data with a sea surface elevation variance of less than 10 cm were discarded (white patches).**

## 3.3 Distinguishing local and regional phenomena

When looking closer at changes in tidal characteristics from both SLR and bathymetry evolution (Table 1, scenario 4 minus 1) in the inner German Bight (Figure 4), phenomena can be distinguished by scale into a local (e.g., tidal basins, estuaries)





and regional (e.g., southern North Sea) component. We observed that the regional changes in tidal high water (Figure 4, a)
either exceeded, fell behind, or adopted the modeled SLR magnitude at the coast. Similar observations were made in tidal
low water (Figure 4, b). Here, the combined changes from SLR and bathymetry evolution caused a noteworthy regional
decline in the Southwest followed by alternating increases and decreases along the counterclockwise path of tidal
propagation. The transition between increases and decreases in tidal high and low water were surprisingly sudden, i.e.,
within kilometers or less.

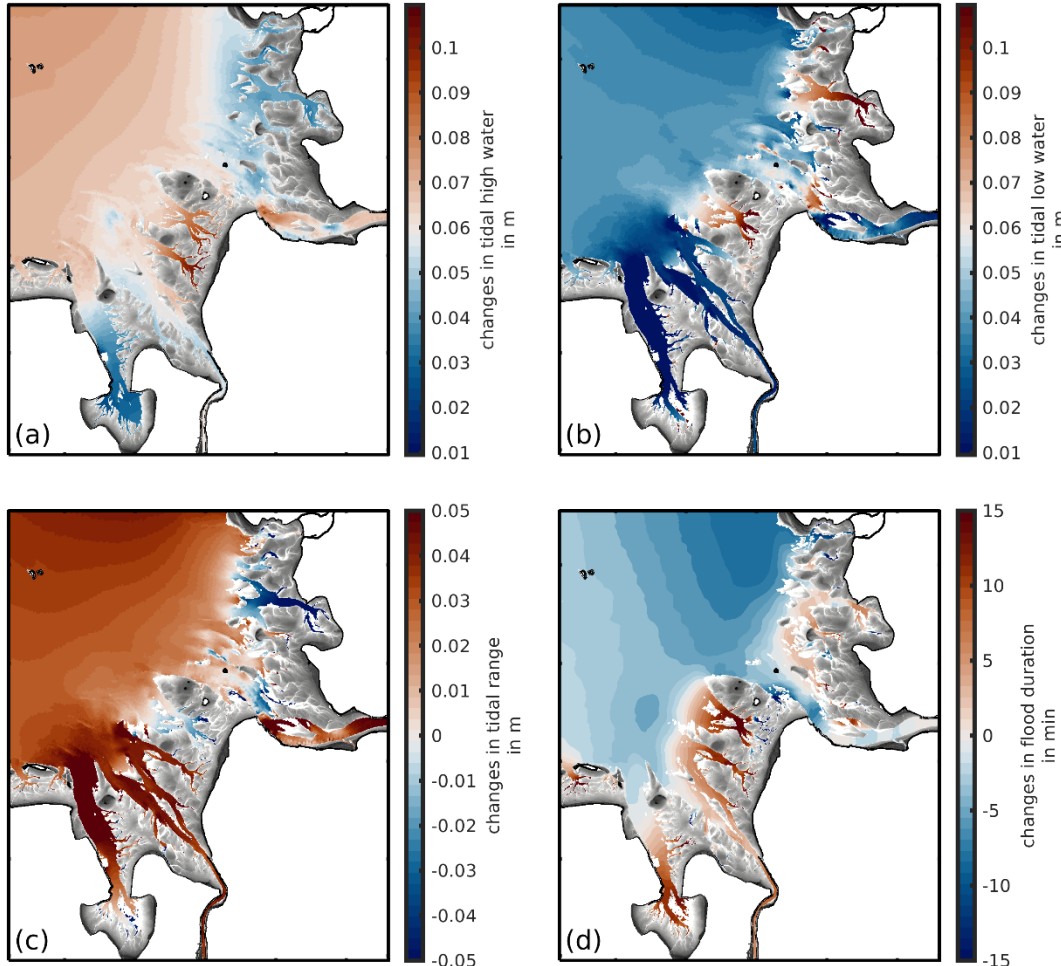

**Figure 4: A detailed view at the changes in tidal high water (a), low water (b), tidal range (c), and flood duration (d) in the inner German Bight (Figure 1, units 9 to 21). The displayed differences result from bathymetry evolution and SLR. Red and blue patches indicate an increase and a decrease of the respective tidal characteristic. White patches represent no-data values or land while gray patches indicate the intertidal zone at low water. Note that tidal high and tidal low water differences were illustrated**
**with respect to the modeled SLR of 5.9 cm.**

Another complex pattern was found when combining the spatially-varying regional and local responses of the tidal range
(Figure 4, c). We observed a regional increase at the outer shoals with noteworthy amplification in the South, contrasted by
local declines in the Southeast and Northeast. Again, sharp increase-decrease transitions (and vice-versa) occurred. Changes



in flood duration (Figure 4, d) further underlined different local and regional phenomena. The regional decline in flood duration turned into an increase at nearly all estuaries and tidal basins as previously identified by Hagen et al. (2022). Our results emphasize that even seemingly small changes in basin bathymetry and geometry can cause local and regional changes in tidal characteristic patterns.

## 3.4 Exploring potential mechanisms

With the nonlinear implications from the bathymetry evolution scenario in mind, we directed our attention towards the relevant mechanisms as expressed by water volume, momentum, dissipation, and the volumetric flow rate. For this purpose, we evaluated the mean energy flux as a measure for pressure gradient-induced momentum, the mean energy divergence as indicator for dissipation, and the mean water flux as estimator of the volumetric flow rate. Given the proportional response of SLR to tidal high and low water, we assessed the changes from bathymetry evolution (Table 1, scenario 2) only in this section. Kinetic energy was negligible in comparison to the pressure-gradient induced momentum. Fluxes were computed over the seaward transect and the cumulative dissipation across the morphological unit's area, respectively (Figure 1, b). Fluxes were described by the mean flood water flux, by the cumulative energy flux, and then normalized by the seaward cross-sectional area during high water. The energy divergence (i.e., dissipation) was standardized using the morphological unit's total area. We chose to correlate the aggregated subtidal and intertidal water volumes over bathymetry parameters (from Sect. 3.1) as the direct implications of mean height changes were offset by lateral intertidal expansion. The subtidal and intertidal water volume were normalized using the respective subtidal and intertidal area of the 1997 simulation. The linear Pearson correlation matrix and a t-statistic were calculated to identify potential relationships. Subsequent relationships were described with R<0.4 for a weak, 0.4≤R<0.7 for a moderate, and R≥0.7 for a strong correlation. It should be noted that linear correlation and t-statistic are only indicative of a relationship considering the small number of samples.

The changes of the sum of all fluxes indicated that SLR increased the mean flood water flux by 1.8 % while bathymetry evolution without SLR decreased it by -8.9 %. This decline was reflected in the tidal energy flux: SLR increased the cumulative mean energy flux relative to the 1997 simulation period (i.e., 2,000 MW) by 1.3% while bathymetry evolution decreased it by -4.3 %. Changes in dissipation (total: 4.4 MW) followed this pattern with a 1.4 % increase from SLR and a -6.5 % decline from bathymetry evolution. The relationship between the parameters was explored further by comparing the discrete distributions in a linear correlation matrix (Figure 5).

Here, decreases in intertidal water volume were shown to be moderately correlated with decreases in energy dissipation. A similar observation was made for decreasing subtidal channel volume which correlated moderately with decreasing flood water flux and cumulative energy flux. Decreases in subtidal water volume were mainly connected to decreasing flood water flux and cumulative energy flux while decreases in intertidal volume hampered dissipation. No relationship was noted between changes in subtidal and intertidal water volume even though most units ceased water storage in both littorals. The decreasing energy flux correlated strongly with declining energy dissipation. Further moderate correlations of the cumulative energy flux were found with subtidal volume increases, and flood water flux decreases. As could be expected, less





dissipation was moderately correlated with intertidal storage loss and with decreases in flood water flux. The correlation assessment therefore suggests that predominantly subtidal and intertidal water volume losses decreased the flood water flux which in turn reduced the tidal energy import. One consequence was the diminishing tidal energy dissipation at the coast that 275 can then be re-connected to the observed regional changes in tidal characteristics such as regionally increasing tidal range.

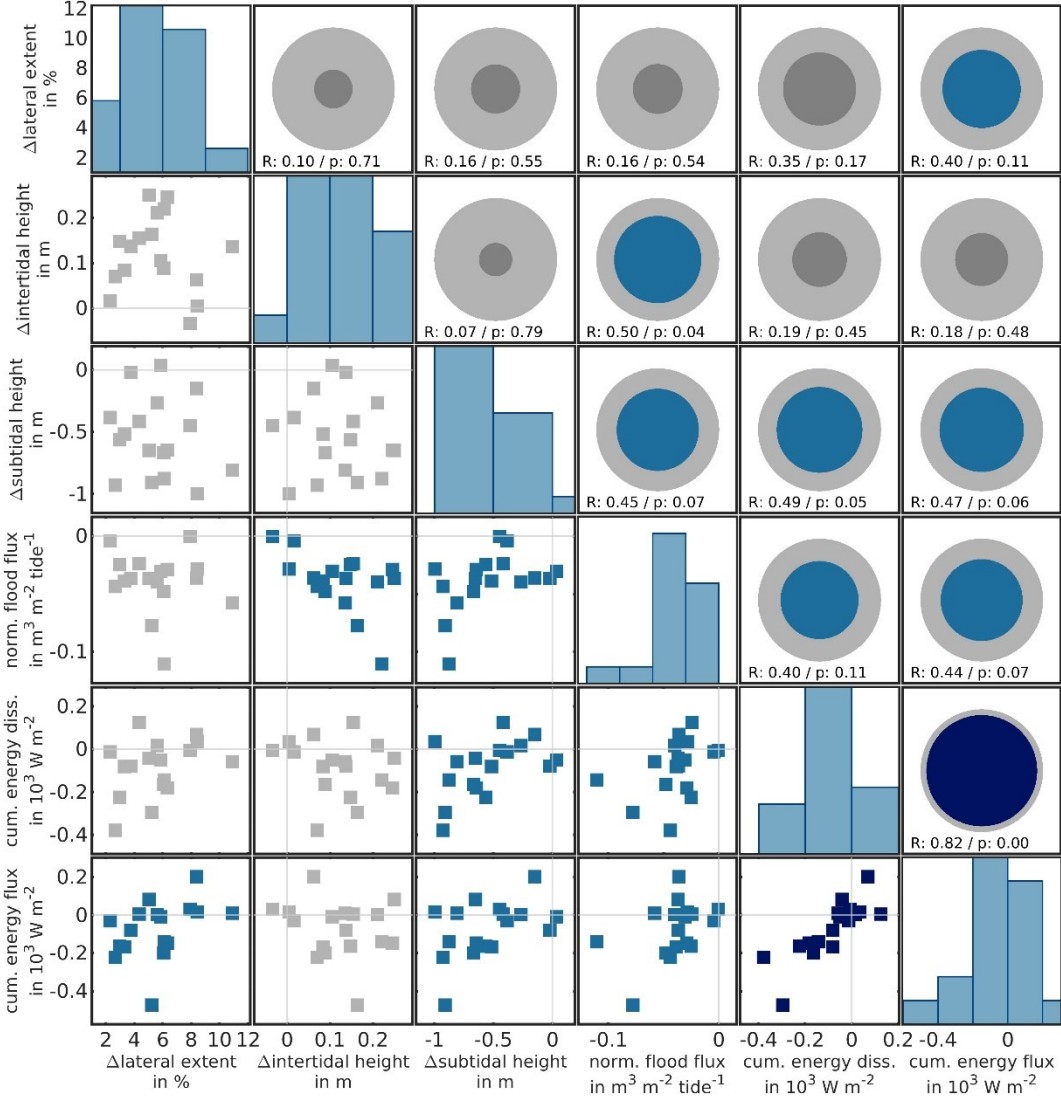

**Figure 5: Graphical representation (described top-down, left-right) of the explorative correlation matrix derived from changes in intertidal water volume, subtidal water volume, normalized flood flux, cumulative energy divergence, and the mean energy flux. The scatter plots with rectangle markers are a graphical representation of the discrete samples in the morphological units.**
280 **Histograms represent the sample distribution qualitatively with respect to the x-axis. The circles indicate the absolute linear correlation coefficient with the gray background circle representing perfect correlation R=1. Discrete correlation value (R) and the p-value of a t-statistic are indicated below the circle. Dark blue colors indicate a strong (R≥0.7), light blue a moderate (0.7>R≥0.4), and gray colors a weak (R<0.4) relationship.**





## 4 Discussion

We investigated the impact of observed SLR and natural bathymetry changes on tidal characteristics for the Wadden Sea. This region comprises vast, non-vegetated intertidal areas and is especially susceptible to changes in physical processes due to SLR, e.g., different patterns of wetting and drying. To avoid bias from the nodal cycle, we investigated tidal dynamics at the diurnal nodal minimum (1997 to 2015). Within a numerical model, MSL and bathymetry evolution were adjusted separately to isolate drivers and effects. The phenomena were assessed by analyzing and correlating changes in bulk bathymetry parameters, energy dissipation, and water- and energy fluxes. We found five noteworthy phenomena: (1) Without bathymetry evolution, tidal high and low water increase proportional to short-term SLR; (2) SLR alone barely affects tidal range and tidal asymmetry; (3) bathymetry evolution affects all tidal characteristics locally and regionally - often counterbalancing the effects of SLR; (4) local and regional changes in tidal characteristics do not necessarily correspond when SLR and bathymetry evolution coincide; and (5) the decline in flood water- and energy flux from bathymetry evolution decreases energy dissipation. This decrease was one order of magnitude higher than the increases from SLR.

### 4.1 Implications for observational tidal gauge trend assessments

Our modeled changes in tidal characteristics were in line with long-term analysis of measured data. A tidal gauge assessment in the period of 1954 to 2014 for tidal range, high, and low water established similar spatial patterns with larger increases in tidal high water and tidal range in the Northeast of the German Bight, lower rates in the West and Southeast, and local effects near land reclamation zones (Ebener et al., 2020; Hagen et al., 2022; Jänicke, 2021). More importantly, a proportional relationship between SLR and tidal high water and a weaker relationship to the tidal low water was established. The latter weak correlation was suspected to be masked by coastal bathymetry evolution or other unknown effects (Jänicke, 2021). This assumption could be corroborated by our findings (Figure 3, d-f) which have shown that the tidal low water does also respond proportionally to sole SLR while bathymetry evolution led to major local and regional shifts (see also Figure 4, b). Our findings therefore present evidence that the proportional relationship between SLR and tidal low water is a reasonable assumption in principle although it can be overshadowed by coastal morphodynamics. Sometimes, the local decrease in tidal low water from bathymetry evolution even compensated the proportional increase from SLR (i.e., Figure 4, b). As this effect cannot be distinguished using only observational data or coarse global or regional models with little grid resolution near the coast, our work highlights that the impact of coastal bathymetry evolution under SLR may locally be larger than previously suspected. The identification of statistically significant changes in tidal characteristics from tidal gauge must therefore be related to coastal bathymetry evolution.

### 4.2 The combined effect of SLR and bathymetry evolution

A detailed view at the coast established that the local and regional response can alternate sign within kilometers (Figure 4). Our results support Jacob and Stanev (2021) who found that the tidal response of the Wadden Sea to bathymetry evolution



was comparable to the implications of SLR. While the regional consequences of bathymetry evolution of our model corresponded with the established behavior of tidal constituent amplitude and phase under SLR (Jacob et al., 2016; Jacob and Stanev, 2021), the focus on effects near the coast provides novel insights. The compensational effect of intertidal accretion on regional tidal characteristics under SLR was previously noted in studies with artificial channel deepening and intertidal accretion (Wachler et al., 2020; Jordan et al., 2021) but none of the latter considered the possibility of lateral intertidal expansion and accretion outpacing SLR. Figure 3 and Figure 4 present strong indications that tidal characteristics are a function of MSL and coastal bathymetry leading to both local and regional effects. These can either negate or amplify the local implications of SLR. Our work connected these dots by corroborating changes in tidal characteristics from SLR and bathymetry evolution with the relevant mechanisms, i.e., volume, flux, and momentum. Decreasing tidal prism from bathymetry evolution, as also noted Jacob and Stanev (2021), reduced mean tidal energy which in turn weakened absolute energy dissipation at our study site (Figure 5). Therefore, the popular assumption that dissipation increases under SLR (Rasquin et al., 2020; Jordan et al., 2021; Holleman and Stacey, 2014; Lee et al., 2017) does not necessarily apply. If SLR alone was considered, however, our results corresponded well to previously mentioned studies and other SLR-impact assessments in the German Bight (Wachler et al., 2020) with more dissipation. Our work therefore adds the important detail that bathymetry evolution of channel-shoal environments is not always proportional to MSL or tidal range contrary to previous studies (Hofstede, 2015; Hofstede et al., 2018). This can be seen in Figure 2 (d) with only few observed increases in mean intertidal height being comparable to SLR.

### 4.3 Suspected feedback mechanisms between energy transport and bathymetry evolution

One motivation for our study was that the intertidal accretion surpassed SLR in large parts of the Wadden Sea (Figure 2, d) as first discovered by Benninghoff and Winter (2019). Our model gave an indication on how nonlinear feedback processes made this accretion surplus possible, assuming that the Wadden Sea strives towards dynamical equilibrium.

The linear scaling of tidal high and low water with SLR leads to a sediment deficit. We believe that this deficit triggers feedback because sediment must be imported or moved to re-establish equilibrium. Lateral intertidal expansion (Figure 2, e) could be one consequence that had the side effect of reducing the cross-sectional and surface area of tidal channels (Jacob and Stanev, 2021; Lepper, 2023). The resulting lower hydraulic capacity and resulting flood water flux leads to a decline in energy influx (Figure 5). Lower mean tidal energy influx decreases local tidal amplitude resulting in negative feedback (e.g., Figure 4, c), which is in line with the declining mean flood and ebb flow velocity noted by Hagen et al. (2022). Consequently, lower tidal energy import weakens sediment resuspension and energy dissipation. Less local dissipation draws less energy of the tidal wave along the path of tidal propagation. The consequence is that tidal energy can be transported eastward with less resistance ultimately increasing tidal range in the eastern German Bight: Regional tidal amplification from reduced local dissipation presents as positive feedback (e.g., Figure 3, i). Another feedback mechanism concerning declining flood duration was suspected in Hagen et al. (2022). Lateral intertidal expansion, in other words subtidal retreat, increased the relative intertidal storage and decreased the tidal channel volume to weaken flood dominance (Friedrichs, 2010; Lepper,



2023). This mechanism presents a negative feedback because locally decreasing flood dominance hampers sediment import into the coastal zone.

Concluding, our modeling study and the evaluation of mechanisms suggests the following framework for feedback mechanisms between tidal and bathymetry evolution under SLR: The sediment deficit from SLR disrupts the dynamical equilibrium and triggers bathymetry evolution and local changes in forcing (i.e., tides and waves). Any changes in forcing, however, initiate further local feedback which again depends on local features such as sediment availability or a soils resistance to erosion. This chain of processes is permanently amplified by SLR as the sediment deficit increases proportionally. Mark that bathymetry evolution in this matter does not concern natural channel migration or other natural coastal dynamics that may be expected within dynamical equilibrium. We furthermore recognize that several other potential geomorphological (e.g., grain-grain interaction) and baroclinic feedback mechanisms are likely at play. We recommend future research using conceptual modeling to enlighten the magnitude of feedback mechanisms as our work clearly shows that their understanding is imperative to project the development of tides and coastal morphology with confidence.

## 4.4 Limitations and uncertainty

Our study design regarded SLR and bathymetry evolution separately and assumed all baroclinic processes to remain constant during our study period. This is a limitation to our experimental design as coastal morphodynamics are the consequence of several dependent and independent processes such as tides, waves, estuarine sediment trapping, biological potential, sediment availability, and sediment properties. We believe constant external forcing to be a reasonable simplification during our study period because long-term modeling and observational data analysis indicated insignificant changes in median and peak wind velocity (Krieger et al., 2021), peak sea surface elevation, and median and peak wave heights (Lepper, 2023). Furthermore, increase-decrease patterns were corroborated by observational data analysis (Sect. 4.1). It should be noted that baroclinic effects such as ocean warming may also play a role at our shallow study site concerning changes in tidal high and low water (Jänicke et al., 2020).

Despite this simplification, identifying mechanisms, drivers, and effects remained cumbersome. A modeling approach in complex systems bears the problem that cumulative effects from local and regional dependencies exist. Local shallow water effects modify and distort the passing tidal wave which then affects all subsequent basins and estuaries (Dronkers, 1986). The Wadden Sea and other channel-shoal environments are sediment-sharing systems (Sect. 2.1) and our studies indicate that this statement extends to the tidal energy transport. The discussion of mechanisms can thus not be described as conclusive as they would be in conceptual models or in theoretical considerations. Anthropogenic measures such as sediment management and estuarine realignment add to this effect. We suggest that future research should conduct conceptual modeling based on our findings to further disentangle drivers and effects that affect the morphological adaptation of our coasts to SLR. Another interesting aspect is the investigation of cumulative effects in other channel-shoal environments for a better understanding of regional phenomena.





## 5 Conclusion

The suggested relationship between SLR and bathymetry evolution under SLR affects the applicability of large-scale numerical models without appropriate downscaling when estimating the local and regional consequences of SLR. Not only is a high model resolution inevitable to resolve coastal bathymetry, but also a reasonable assumption for bathymetry evolution

under SLR must be made when conducting SLR impact modeling. An inaccurate representation of coastal morphology or artificial projections of future morphological changes were previously shown to easily change the sign of a tidal amplitude prediction near the coast (Pelling et al., 2013b; Jordan et al., 2021) which can make any SLR impact studies ambiguous. Our study stresses this limitation because even seemingly minor coastal bathymetry changes had local to basin-wide effects with local effects being either one order of magnitude larger or compensational. This can also have a profound impact on

observational gauge data assessments as most tidal gauges are located near the coast. We suggest to encounter this uncertainty with ensemble simulations with ensembles for bathymetry evolution under SLR, similar to Jordan et al. (2021) to develop more robust SLR-impact predictions in the future.



**Code availability**

Not applicable

**Data availability**

The authors used the commercial hydrodynamic numerical modeling software UnTRIM² (https://wiki.baw.de/en/index.php/UNTRIM2, available on request). Input data for the model, as disclosed and cited in Section 2.2, can be accessed as follows: Vaklodingen:

https://publicwiki.deltares.nl/display/OET/Dataset+documentation+Vaklodingen; COSMO-REA: https://reanalysis.meteo.uni-bonn.de/?COSMO-REA6; FES2014b: https://www.aviso.altimetry.fr/en/data/products/auxiliary-products/global-tide-fes.html. Bathymetry data were adapted from the public EasyGSH-DB data collection (Sievers et al., 2020). Tidal characteristic analyses were conducted with our access-restricted NCANALYSE framework (https://wiki.baw.de/en/index.php/NCANALYSE) as described in Section 2.4. Tidal data at the gauge "Helgoland" is

available upon request at the German Water and Shipping administration WSV. We used the EMODnet bathymetry products, related to MSL, for modeling and as background in the top left of Fig. 1 and within our numerical model (EMODnet Bathymetry Consortium, 2018).

**Interactive computing environment**

Not applicable.

**Sample availability**

Not applicable.

**Author contribution**

- Conceptualization: RL, FK
- Data curation: RL
- Formal analysis: RL
- Funding acquisition: FK
- Investigation: RL
- Methodology: RL
- Project administration, RL, FK



•     Resources: LJ, IH

      •     Software: RL

      •     Supervision, FK

      •     Validation: RL, IH, LJ, CJ

      •     Visualization: RL

•     Writing (original draft): RL

      •     Writing (review and editing): IH, LJ, CJ, FK

## Competing interests

The authors declare that they have no conflict of interest.

## Disclaimer

Not applicable.

## Acknowledgements

This research was part of the TrilaWatt project, funded by German Federal Ministry of Digital and Transport (BMDV) together with the generous mFUND initiative, grant number 19F2206-A and the BMDV "network of experts" group. C. J. was supported by the DAM-SN Coastal futures project (03F9811G) funded by the German Federal Ministry of Education

and Research (BMBF).

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
