# Peer review of "Exploring the Tidal Response to Bathymetry Evolution and Present- Day Sea Level Rise in a Channel-Shoal Environment"

_EGUsphere, 2024_

## Author Comment (AC1)

Dear anonymous referee #1,

thank you for the elaborate review of our manuscript. We appreciate your time and effort that went into reading, understanding, and improving our work. We are certain to profit from your remarks. Below, we answer to your concerns point-by-point. Please note that we included a false version of the correlation matrix in Fig. 5 in the original manuscript. The text already referred to the correct parameters and the conclusions remain unchanged. We have corrected this mistake in the revised manuscript.

Sincerely,
Robert Lepper (on behalf of the authors)

**Answer to referee #1**

[…] However, there are a few obscure aspects which it would be good if they could be further elaborated or commented in the manuscript to strengthen better the validity of the methodology. These are summarized as follows:

**Major points**

1) The basic assumption in lines 123-125 that SLR is the dominant driver because the external forcing parameters are constant neglects the non-linearities between them which, even though constant, might cause effects surpassing that of SLR and features that could be present even without the inclusion of SLR.

   *Answer: Our study was carried out as a "what-if" study. One model run investigated the mean sea level change of the period 1997 to 2015 prescribed as a constant offset at the open boundary of our model. As you noted correctly, this approach neglects non-linearities in nature that affect mean sea level. However, the goal of our "what-if" study was to investigate the impact of net mean sea level change rather the resulting non-linear implications from it. We believe that this simplification was necessary to retain the chance to interpret model results but we are aware now that further clarification of our approach was needed.*

   *Action: We clarified the "what-if" approach and information about the observed SLR at our open boundary in Sect. 2.3. of the original manuscript. The pointed-out limitation to our study's design considering non-linear phenomena was acknowledged in Sect. 4.4.*

2) The 10.6 cm RMSE for high waters is higher than the prescribed SLR of 5.9cm. A comment is needed to justify that the extracted conclusions on the role of the SLR are not affected by this difference.

   *Answer: We agree that the deviation of modeling results and observations for tidal characteristic values and the systematic, constant change of mean water level needs further explanation in the manuscript. The mean RMSE of tidal high water in model validation was mentioned as an indicator for the successful model validation. The departure of modeling results from observations does not affect the results of our "what-if" study as runs were conducted with identical calibration and within an identical period for all sensitivity runs.*

*Action: We clarified in L101 of the original manuscript why this deviation of modeling results and observations has no impact on our results.*

3) It is not clear from the manuscript if the sea level rise is implemented as a uniform increase over the entire domain. If this is the case, then the model carries out the same calculations but over uniformly increased depths while the depth gradient which is directly involved in the flow calculations, remains unchanged. As a result, the HW and LW increase is on a similar level as it is depicted in Fig. 3(a) and 3(d) where minor differences can be observed between them. So, this means that the conclusion would be similar irrespective of the implemented SLR.

*Answer: Thank you for this remark. Sea level rise was implemented uniform across the domain (as noted in remark 1) which is an important assumption we must mention in the paper.*
*Even though the gradient in bathymetry stays the same in the MSL-only scenario, different wetting and drying behavior in the intertidal zone does affect tidal dynamics when increasing MSL. This can be seen more clearly at larger amplitudes of SLR (e.g., in Jordan et al., 2021) but was also noticeable in our study: Some degree of nonlinearity may be observed in Fig. 3 (a, d) when looking closely at the Eastern tidal basins.*

*Action: We acknowledged in remark 1) that SLR was applied uniform across the model's domain. In addition, we added a statement in Sect. 4.4. that the bathymetry and it´s gradient stays the same while mean sea level increase alters wetting and drying patterns.*

4) The authors imply in line 163 that the 2015 bathymetry includes already changes due to SLR but then mention that these were negligible and included. Which of the two is right? Was this a design scenario decision to not include changes due to SLR or were these investigated and then found negligible?

*Answer: We understand now that the two first sentences of 3.1 were misleading. We did use different bathymetries to estimate bathymetry parameters for 1997 and 2015 and in these the effect of sea level was considered. The second sentence was intended to clarify that changes in bathymetry parameters (e.g., intertidal area) can arise from changes in mean sea level alone. By ways of example: If the MSL rises by several meters at constant bathymetry, the intertidal area diminishes and the mean intertidal height would change. We investigated this effect by computing the same bathymetry parameters of the bathymetry of 1997 using the MSL of 2015 but found that these changes were much smaller than the phenomena described in Fig. 4 which is why we added this sentence.*

*Action: We conclude that this sentence in L162/163 of the original manuscript adds no value but adds confusion. We decided to remove the sentence.*

**Minor corrections**

a) Line 38 It is either sea surface level or sea surface elevation and not height. *We are certain that the term sea surface height (SSH) is a common term used in our field. We have unified all references to the sea surface height to "sea surface height" for more clarity.*

b) Line 83 Write "approximately" and not "approx." *Action: Changed in the manuscript.*

c) Line 103 Delete the second (a) in the line. *Changed in the manuscript.*

d) Line 126 It would be good to add the Helgoland sea level gauge location on Figure 1 map. *We added the gauge's location with a red triangle. (no track changes). The reference in the text was adjusted accordingly.*

e) Line 135 As above, sea surface elevation and not height. *See a)*

f) Line 162 It is scenario (2) and not scenario (4). *Changed in the manuscript*

g) Line 168 Better to use 'note' instead of 'mark'. *Changed in the manuscript.*

h) Line 200 I think you mean constant or amplified tidal low water instead of range. *Changed in the manuscript.*

i) Line 197 "on the contrary" instead of "contrary". *Changed in the manuscript.*

j) Line 199 "on" instead of "with". *Changed in the manuscript.*

k) Line 325, 'by' Jacob and Stanev (2021). *Changed in the manuscript.*

l) Line 384 a "high resolution model". *Changed in the manuscript.*

**Figures**

m) Figure 1 (a) occurs twice in the first sentence of the legend. Please delete one of the two. – *changed in the manuscript* – Please make thicker the depth contours or at least more visible and add the depth indication on the map. *Depth contours were changed to a lighter gray and their depth was indicated.*

n) Figure 2 'heights (m NHN)' instead of 'heights in mNHN' on the horizontal axis. – *we have internal guidelines on how to format units in graphs. We wish not to deviate from these guidelines. We will adjust the units, however, if the second reviewer or the editor agree with your remark* – The distinction between the 1997 and 2015 pdf is not visible. Please use another color. *We made the colors darker and thickened the outline of each to help distinguish between the colors (no track changes). The colormap was adjusted to another red-blue scheme (crameri, vik) for more saturated colors.*

---

## Author Comment (AC2)

Dear anonymous referee #2,

thank you for the thorough review of our manuscript. We appreciate that you took the time to read our work in depth and made remarks that will improve the quality of it. Below, we answer to your revisions point-by-point. Please note that we included a false version of the correlation matrix in Fig. 5 in the original manuscript. The text already referred to the correct parameters and the conclusions remain unchanged. We have corrected this mistake in the revised manuscript.

Sincerely,
Robert Lepper (on behalf of the authors)

**Answer to referee #2**

**Major points**

1) I miss a mechanistic explanation for the results, at least within the results sections. There are some attempts in the discussion but this is mainly focused on the evolution of the channels and sediment transport capacity. Section 4.3 points to some possibilities, but this is at the local scale. What drives the large-scale changes in the tidal prism? Comments on this, or even a speculation about mechanisms is needed.

*Answer: Thank you for raising this question. A mechanistical speculation about the local changes in tidal prism was given in section 4.3. as you already pointed out. We would like to refer to Sect. 4.2. where a mechanistic explanation considering tidal prism was debated. We do agree that some additional information is necessary in the result chapters.*
*In our opinion, local and regional phenomena must be separated more in future research, because their combination complicated the distinct identification of mechanisms. If we were regarding an isolated tidal basin, a mechanistic explanation would be simpler, but the cumulative effect of multiple local distortions made the separation of drivers and effects cumbersome. For this reason, we suggested future research for a better understanding of the cumulative effect at the end of Sect. 4.4.*
*Still, we believe that further possible explanations are possible and necessary at the end of Section 3.4. (Fig. 5) and have elaborated the end of this section to explain for the observed correlation (or the lack thereof).*

*Action: We elaborated the end of Sect. 3.4. and included a reasonable chain of indications for the observed behavior by clearly dividing the behavior to a coastal and to a shelf-sea context.*

**Minor points**

a) L56-57: please use the same units for SLR and accretion – cm yr-1 would be preferred. Also, please refer to Figure 1 here. *Changed in the manuscript to cm yr-1.*

b) L100: how do these errors propagate through the solutions when you change bathymetry? They are larger than your SLR signals later on. This must be commented on.

*Answer: See "answer to reviewer #1" point 2). Validation of the model was carried out for all years in the period of 1997 to 2015 (Hagen et al., 2021). Mentioned errors are an average of the RMSE of each year at each gauge. We included an excerpt of the respective RMSEs (1997/2015) here to demonstrate comparability:*

| gauge | high water | tidal range | flood duration |
|-------|-----------|-------------|----------------|
| ALW | 9/6 cm | 15/12 cm | 15/15 cm |
| DWG | 14/10 cm | 15/13 cm | 14/9 min |
| BAL | 19/14 cm | 16/16 cm | 13/14 min |
| CUX | 14/14 cm | 15/20 cm | 14/13 min |

*Action: We clarified this matter following the remark of reviewer #1. No further action was taken.*

c)  Mark -> Note. Also, on L356. *Changed in the manuscript.*

d)  L221-223: Distinguished -> separated may be a better word. *Changed in the manuscript.*

e)  L326: Is this a popular assumption? In my mind it is the opposite and SLR leads to weaker dissipation. I guess this depends very much on local conditions, so maybe this could be re-worded and you can state that you see a decreased dissipation, as do several other papers (including global studies) whereas others show increases. This leads to an important conclusion: regional responses must be modelled explicitly, and one region is a poor representation for general signal in another region.

*Answer: In our opinion, this point is tightly linked with your major comment and our action is to be regarded in context. We understand now, that this statement is confusing from an oceanographic point of view. From our coastal perspective, it was popular belief that more extensive tidal flats increase overall dissipation near shore and we were surprised to find out that the opposite was the case. Your conclusion is essential and we have included it in the manuscript.*

*Action: We reworded the statement to what it was: A citation of the authors from L327 (of the original manuscript) and added your conclusion to Sect. 5.*

---

## Author Response (AR1)

Dear John Huthnance,

we have responded to all referee's comments in the open discussion section. We would like to thank you for the swift coordination of the review and are very pleased with the process at Ocean Science. All changes, aside from the exchanged figures, were made using the track-changes functionality. We hope you are satisfied with our corrections. Please contact us if there are any further changes or remarks that need clarification.

Sincerely,
Robert Lepper (on behalf of the authors)